# Community engagement in health services research on elimination of lymphatic filariasis: A systematic review

**Cho Naing** [1]*, **Norah Htet Htet** [2], **Htar Htar Aung** [2], **Maxine A. Whittaker** [1]*

**1** Division of Tropical Health and Medicine, James Cook University, Queensland, Australia, **2** School of Medicine, International Medical University, Kuala Lumpur, Malaysia

☯ These authors contributed equally to this work.
* cho3699@gmail.com (CN); maxine.whittaker@jcu.edu.au (MAW)

**Data Availability Statement:** All relevant data are within the manuscript and its Supporting Information files.

## Abstract

This study aimed to contextualize the extent, nature, and quality of community engagement in health services research on eliminating lymphatic filariasis in low-and middle-income countries of Southeast Asia and Pacific Region. We performed a systematic review, and the results were reported according to the PRISMA-S checklist. Relevant studies were searched in health-related electronic databases, and selected according to the inclusion criteria. Sixteen studies with various study designs were identified. The majority (68%) were conducted in India. Lay people, community leaders, and volunteers were the most common groups of community members (12/16,75%). Overall, the majority (13/16, 81%) were at the 'moderate level' of engagement in research context mainly by 'collaboration' in 'developing methodology' 'collaboration' in data collection and 'collaboration' for 'dissemination of findings. The common barriers to the community engagement were lack of involvement of participating bodies and technology-related issues. In conclusion, the insufficient description of the community engagement process in the studies limits a deeper understanding and analysis of the issue. Future well-designed prospective studies with attention to the description of mechanisms of engagement, facilitating the whole process and reporting the community level outcome are recommended.

## Introduction

Infectious diseases pertinent to Neglected tropical diseases (NTD) are a broad group of communicable diseases that have proliferated in tropical and subtropical climates, mainly across low- and middle-income countries (LMICs) [1]. Although not fatal, WHO has ranked lymphatic filariasis, which is an NTD, as one of the world's leading causes of permanent and long-term disability [2]. In 1997, the World Health Assembly endorsed lymphatic filariasis for global elimination of infection as a public health problem [3]. In order to achieve the ultimate goal of interrupted transmission to eliminate lymphatic filariasis from public health problem, WHO recommended the treatment of the total population in

**Funding:** This work received financial support from TDR (the Special Programme for Research and Training in Tropical Diseases, co-sponsored by UNICEF, UNDP, the World Bank and WHO) [Project ID AP21-00287]. CN, MAW, NHH, and HHA received this award. None of the authors received salary or other funding from commercial companies. None of these authors work at commercial companies. The funders had no role in study design, data collection and analysis, decision to publish, or preparation of the manuscript.

**Competing interests:** The authors have declared that no competing interests exist.

endemic areas for at least 5–7 consecutive years through annual or biannual mass drug administration (MDA) [2, 4].

Lymphatic filariasis prevalence substantially declined from 2000 to 2018, but it appears that not all areas will achieve the global programme to eliminate lymphatic filariasis (GPELF) targets by the original goal of 2020 [4]. In 2020, WHO estimated that 863 million people in 50 countries were living in areas that require preventive chemotherapy to stop the spread of infection [5]. In reality, the health systems of most endemic countries may be challenged to achieve this target of MDA due to many reasons [6], including local perceptions of health needs and the support of local people [6, 7]. Many health services, particularly in developing countries, function on the basis of limited resources. Hence, community's engagement/involvement can be a means of making more resources available by drawing upon local knowledge and resources to complement what is provided by the formal health services [7].

From the health services perspective on the research context, studies reported that community engagement (CE) strengthened the conduct of research [8, 9], and was a core value in participatory health research [10]. CE is increasingly promoted in health services research (HSR), the concept itself, and how it is best implemented in practice, is understudied and contested [10]. HSR refers to a multidisciplinary field of inquiry, both basic and applied, that examines access to, and the use, costs, quality, delivery, organization, financing, and outcomes of health care services to produce new knowledge about the structure, processes, and effects of health services for individuals and populations [11]. Numerous reviews have focused on CE in general health research [12–14], in which process outcome or health outcomes were addressed. In aligned with the burden of NTDs, this study focused on lymphatic filariasis and confined to low- and middle-income countries (LMICs) in Southeast Asia (SEA) and Pacific Region.

Taken together, we did a systematic review to address a question: What studies are available that have assessed the CE in HSR on elimination of lymphatic filariasis? Hence, our objective was to contextualize the extent, nature and quality of CE in HSR targeted towards the elimination of lymphatic filariasis in LMIC countries of SEA and the Pacific region. Specific objectives were to characterize the types of CE in HSR, describe the mechanisms used by the researchers to engage with communities on HSR, and summarize factors affecting CE in HSR aimed for the elimination of lymphatic filariasis in LMIC countries of SEA and Pacific region.

## Materials and methods

A protocol of this study was approved by the Ethics Review Committee of the International Medical University in Malaysia (Project ID: R 272/2021). We reported the current review adhered to the Preferred Reporting Items for Systematic Reviews and Meta-Analysis (PRISMA) guideline [15] (S1 Checklist).

### Search strategy

A simple database search was performed on PubMed, CINAHL, plus, Google Scholar, PsycInfo, ProQuest Dissertations & Theses: UK and Ireland, Web of Science, Science Direct, Cochrane Methodology Register (CMR) and the WHO library database (WHOLIS), using the keywords with appropriate Bolen operators: "community engagement" "community participatory" "action research" "participatory research" "participatory action research" "community-based research" "filariasis" " lymphatic filariasis" "elephantiasis" Search was limited to studies in English from 1978 and January 2022. A start date of 1978 aligned to the 40th anniversary of the 1978 *Alma Ata declaration*, regarding primary health care [3, 16]. Details on the search strategies are presented in S1 Table.

## Inclusion criteria

We set up inclusion criteria in the PECOS (i.e., participants, exposure, context, outcome, study design) framework (Table 1). In brief, any study which evaluated or reported on CE in health services research (HSR) for lymphatic filariasis, or which reported on CE in an individual study for lymphatic filariasis was eligible for inclusion.

## Exclusion criteria

We excluded studies, which did not explicitly report CE in the HSR context. Studies on CE outside lymphatic filariasis were not included. Also, we excluded diagnostic accuracy studies, non-empirical studies (e.g., commentaries, narrative and systematic reviews), knowledge, attitude and practices (KAP) studies, and drug efficacy studies were excluded.

## Assessment of the methodological quality

We evaluated the methodological quality of the included studies using "Risk Of Bias In Non-randomised Studies-of Interventions" (ROBINS -I) tool [17]. In the ROBINS-I tool, seven types of bias were assessed: bias due to confounding, bias in the selection of participants in a study, bias in the measurement/classification of interventions/exposures, bias due to deviations from intended interventions/exposures, bias due to missing data, bias in the measurement of the outcomes, and bias in the selection of the reported results.

## Assessment of the level of community engagement in HSR

We assessed the degree of good practice of CE. To do so, we identified the level and extent of CE as a continuum of community involvement (leading, collaborating, consulted, informed and not informed/ unclear) [18].

**Table 1. Inclusion criteria.**

| Frame | Description |
|---|---|
| **Population** | Individuals/communities at risk in lymphatic filariasis, regardless of age and gender. |
|  | "Communities" is as defined in the primary studies. |
| **Exposure (E):** | Program/intervention targeting the health services need to address lymphatic filariasis, which involves community/stakeholder involvement or engagement and provides mechanisms and/or processes of community engagement. |
| **Context (C):** | Community-based and/or primary health care in any settings (i.e., rural or urban) of the LMIC in SEA and Pacific region. |
| **Outcome (O)** | at least one health outcome (e.g., self-efficacy/self-esteem/ self-regard, beneficial effects or positive trends, clinical or physiological outcomes) and/or process evaluation. Either positive, neutral or negative outcomes were considered. The following list is not exhaustive, but some of these are operationally defined as below. • Beneficial effects: An outcome that is statistically significant in favours the community intervention (i.e., a positive directional effect) • Self-efficacy: the belief in their abilities, specifically their ability to meet the challenges ahead of them and complete a task successfully. • Process evaluation: aiming to understand the functioning of an intervention, by examining implementation, mechanisms and contextual factors. [Please see more details of outcome descriptions in Table 2]. |
|  | We measured these outcomes as frequency/percentage for categorical data, and mean/standard deviation (SD) for continuous data. For skew data, we used median and interquartile. |
|  | If the studies did not assess the outcomes of our review, we narratively described these studies. |
| Study designs | Quantitative and qualitative study designs involving primary data collection. |

In each level, the two investigators (CN, NHH) independently rated across research phases such as 'developing ideas', 'developing methodology', 'data collection/analysis, 'report writing' and 'dissemination. We give 1 (+) score to 'leading' or 'collaborating', while 0 scores for the remaining three attributes, as described elsewhere [12]. Hence, the highest score that a study can be achieved is five. As an example, study AA showed CE as 'consultation' at 'developing ideas', 'collaboration' at 'developing methodology', and 'informed' at data collection/ analysis, but 'not sure' at reporting writing and 'collaboration' at dissemination stage,) achieved a total score of 2 (i.e., 0+1+ 0+0+1 = 2). To determine the extent of CE, the level of engagement across all aspects of the study was summed up and the extent determined as high extent (score 4–5), moderate extent (score 2–3) and low extent (score 0–1).

For the quality assessment as well as the risk of bias assessment, any discrepancy between the two investigators (NHH, CN) were settled by discussion with the third investigator (MAW/HHA).

### Data collection

Two investigators (CN, NHH) independently screened the titles and abstracts yielded from the searches in electronic databases. The two investigators independently retrieved full-text copies that were deemed relevant and checked their eligibility. Full-text copies of all the articles from the search output were able to retrieve. The two investigators independently extracted the relevant data using a piloted data extraction sheet.

Textual data were extracted from the studies as described below.

■ Features of the study setting, i.e., the geographical setting, the social, cultural context,

■ Features of the interventions i.e., type of intervention, how it was delivered, where it was implemented and by whom, funding, technical details and any mechanisms targeted by the intervention,

■ Level of participants i.e. communities, households, individuals, details on age and gender

■ Facilitators/barriers encountered.

We also abstracted data regarding the type of HSR. For ease of data collection, we prepared a description of HSR (Table 2).

Any discrepancy between the two investigators in data collections at any stage was settles by discussion with the third investigator (MAW/HHA).

### Data synthesis

Descriptive statistics were done for the important variables in the studies identified. Outcomes were reported narratively (e.g., factors affecting CE in HSR). In addition, qualitative synthesis including contextual data to explain complex issues and complementing quantitative data by explaining the "why" and "how" behind the "what" [9]. For a map indicating the geographic distribution of studies identified, we use *R* version (4.11) pertinent to "*rworldmap*" package. The data used for mapping and commands are provided in S1 Text.

### Results

Fig 1. shows the PRISMA-S diagram of the study selection process. The initial search returned 1069 citations. After the removal of 143 duplicates and 378 irrelevant studies, 548 were screened through their titles and abstracts. The full texts of 33 studies were obtained for further screening. A total of 16 studies were identified for this review [20–35]. The excluded 17 studies and reasons for their exclusion are provided in S2 Table.

**Table 2. Description of health services research in this review.**

| No. | Category | Subtype | Descriptions |
|---|---|---|---|
| 1 | Evaluating the quality of health services | 1A. Structure of health services | ■ facilities (e.g., hospitals, clinics, primary /health centre)<br>■ personnel<br>■ technology (e.g. checklist, diagnostic tool, treatment algorithm, drugs administered, etc) |
| | | 1B. Process of health services | interactions between the health care providers and patients/communities over time |
| | | 1C. **Outcomes of health services**<br>• Beneficial effects: An outcome that is statistically significant in favours the community intervention<br>• Self-efficacy: the belief their own abilities, specifically their ability to meet the challenges ahead of them and complete a task successfully | ■ patient/community-reported health/functional status (e.g. MDA coverage, MDA compliance)<br>■ satisfaction with health status (e.g QoL)<br>■ satisfaction with services (e.g. drug distribution in MDA)<br>■ costs of health services. |
| 2 | Public health perspective on health services | 2A. Primary | services that prevent LF or delay its onset |
| | | 2B. Secondary | interventions that can reduce the impact of LF morbidity once it occurs and slow its progression (e.g. MDA). |
| | | 2C. Tertiary prevention | rehabilitation for disabilities resulting from LF |

Adapted from [19].

LF: lymphatic filariasis; MDA: Mass drug administration; QoL: quality of life

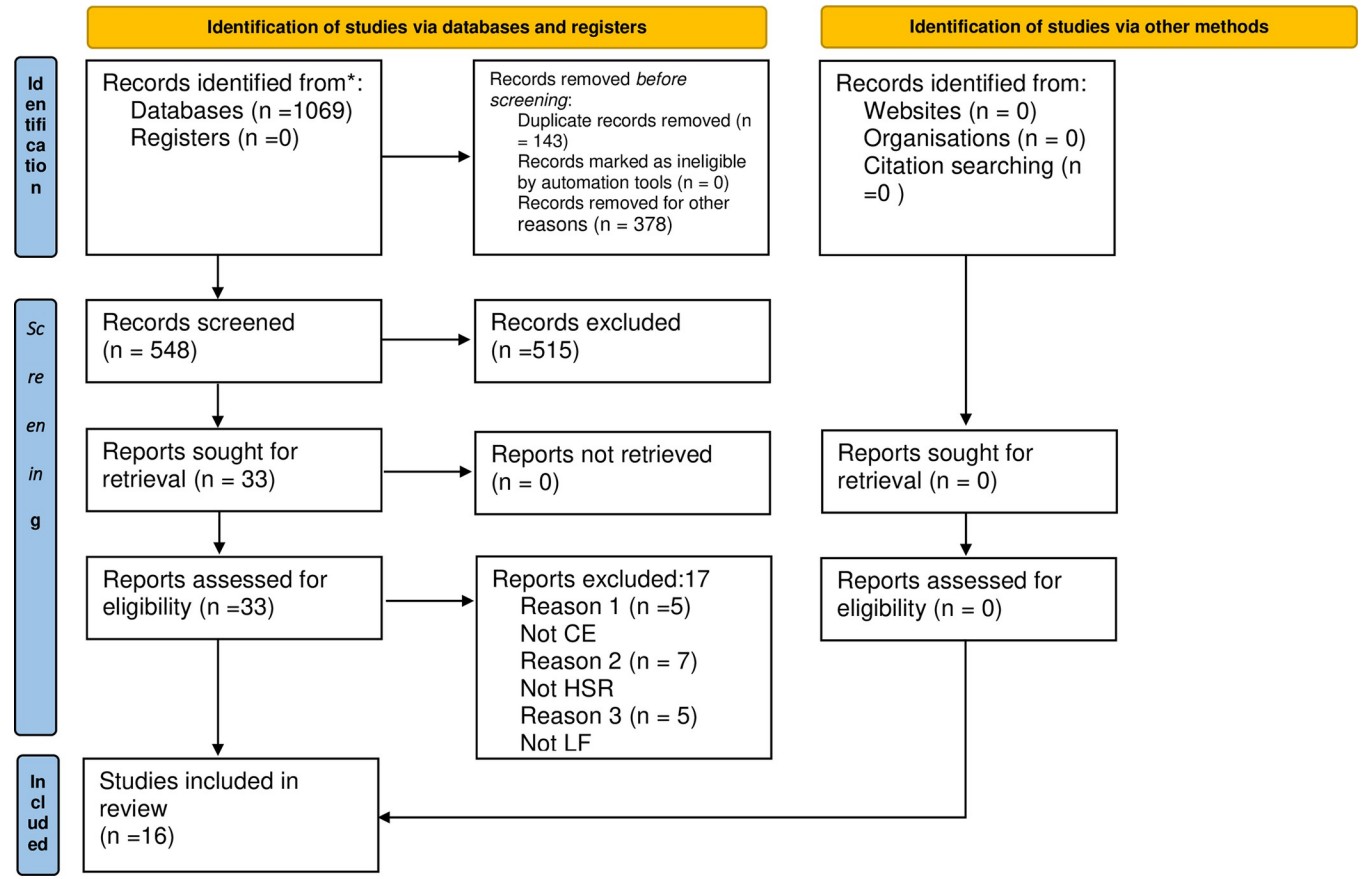

**Fig 1. PRISMA- 2020 study selection process.**

Table 3 presents the main characteristics of studies. There was variation with regards to the country, setting, populations, the process of CE, the measures implemented and the outcomes assessed as well as contextual factors. All 16 studies were published as journal publications and the year of publication spanned from 2000 to 2019. A maximum of two publications were found in 2007, 2013, 2016 and 2018. Studies were largely classified as observational design, but none used the recommended reporting checklist (STROBE:www.strobe-statement.org). Geographic distribution of the included studies across five countries is displayed in Fig 2. The majority of the included studies (11/16, 68%) were conducted in India. There was variation in study designs including community-directed treatment (ComDT) approach design [33], transmission assessment survey method [21], and a longitudinal survey with community empowerment [32]. Of these 16 studies, the majority (13/16, 80%) reported 'outcomes of health services' in the HSR typology. Seven studies (7/16, 44%) reported process evaluations by examining implementation, mechanisms and contextual factors. Only two studies [22, 28] reported three outcomes (i.e structure, process and outcome of HSR) according to HSR typology.

Overall, the majority of studies (14/16, 87%) were with 'moderate risk of bias' (S3 Table).

There were various forms of communities engaged with the health sector staff including lay persons, community leaders, volunteers, faith-based organizations, non-health Government sectors such as academic partners (i.e., teachers, school children/university students), General practitioners (GPs) and non-government organisations (NGOs), among others. Virtual communities were not involved (Table 3).

In the majority of studies (12/16, 75%) the main "form" of community involved in the CE were lay persons, community leaders, and volunteers. Four studies (4/16, 25%) involved more than one form of CE. Only one study [35] included five forms of CE listed in this review (Fig 3).

The extent of CE in various phases of HSR was directly or indirectly measured in a variety of ways across the studies. Overall, 25% of studies (4/16, 25%) [20, 24, 26, 29] were at the low level of engagement in the research context lacking involvement at the stage of 'developing ideas' (i.e. research planning stage) and 'report writing'. The vast majority [21–25, 27, 28, 30–35] (13/16, 81%) were at the 'moderate level' of engagement in research context mainly by 'collaboration' in 'developing methodology' 'collaboration' in data collection and 'collaboration' for 'dissemination of findings' (Fig 3).

Regarding barriers and enablers, there was limited information on these relating to the HSR context, instead, they focussed on the intervention process. The most frequent factors were the lack of involvement of participating bodies and technology-related challenges (S4 Table). On the other hand, the storytelling mechanism [24], and prior communication of the purpose of research to the selected households [26] facilitated CE in the research process.

## Discussion

### Summary of findings

Based on 16 studies across five countries, the present review systematically assessed the extent and level of CE in HSR pertinent to the elimination of lymphatic filariasis in LMICs. Fifteen studies were conducted in the WHO SEA region, and only one study was from the Pacific region. Various forms of the community were engaged in intervention/research reflecting diverse contexts of the locations in which the CE was being undertaken. Less than one-third of the included studies had a 'moderate level' of engagement through collaboration at the stage of 'developing methodology' and 'data collection'. As there was inadequate description of CE in the original papers, there was concern over accuracy in data abstraction. The variation in the mechanism of CE reflected a need for further investigation.

**Table 3. Characteristics of the included studies (N = 16 studies).**

| No. | Study | Country, setting | Design | Type of HSR[#] | Approach |
|---|---|---|---|---|---|
| 1 | Aggithaya, 2013 [20] | India | CP in self-care integrative treatment camp | Process | Assessment of QoL changes in lymphedema following a simplified self-care |
| 2 | Aye, 2018 [21] | Myanmar | Transmission assessment (TAS) method to determine the impact of MDA of prevalence of LF | Outcomes | Each subrural health centre provides health-care services to a cluster of five to ten villages, which have health volunteers & who also assist with the LF MDA activities as community drug distributors. |
| 3 | Babu, 2004 [22] | India | | Structure; Process; Outcomes | The programme is run under management of district MOs through the network of PHCs in rural areas and through municipal health institutions in urban areas. Trained community volunteers & peripheral health workers. |
| | | | | | This project assessed the coverage, compliance, and other operational issues of MDA. |
| | | | | | It involves quantitative surveys on coverage and compliance as well as qualitative survey with FGD approach with community members. The issues identified as community related issues are timely incorporation of IEC materials, active participation by various approaches such as meetings and trainings, different approach between urban and rural population, prioritization of the community and perception by the community, understanding the preventive approach by the community and understanding the side effects of drugs administered. |
| 4 | Babu, 2006 [23] | India | Evaluation of CP in an intervention | Outcomes | drug delivery which involved partnership with stakeholders that include community, for achieving higher compliance in urban MDA. |
| 5 | Dickson, 2018 [24] | Myanmar | Cross-sectional, population-based household survey | Outcomes | A medical student, GP involved additional to Government health staff |
| 6 | Krentel, 2016 [25] | Indonesia | Survey-based micronarrative approach | Process; Outcomes | Survey-based approach using micronarrative approach or a brief history personal experience (social process) with the most recent MDA administration. Factors positively associated for taking LF treatments are also identified. |
| 7 | Lahariya, 2008 [26] | India | Evaluation Qualitative cross-sectional survey | Process; Outcomes | In-depth interviews of the key persons and the community members were used as study tools |
| 8 | Nandha, 2007 [27] | India | Survey for MDA rounds | Outcome | **AWWs** of the ICDS |
| 9 | Narahari, 2013 [28] | India, Gulbarga in Karnataka & Alleppey in Kerala | before-and-after interventional study | Structure; Process; Outcome | A LCT was formed in each center led by an Ayurvedic doctor, a general nurse and midwife, a graduate in medical social work, **ten healthcare assistants (locally recruited personnel** All members were recruited locally. All **patients** were given training in the integrative procedure which involved patient education and the domiciliary protocol. |
| 10 | Patel, 2012 [29] | India, Karnataka | cross-sectional survey | Outcome | Drug distributors were health workers, **AWWs, ASHA and student volunteer** |
| 11 | Ramaiah, 2000 [30] | India | Evaluation of compliance of single dose DEC MDA. | Outcomes | Study on DEC distribution and compliance. Quantitative survey and qualitative assessment of compliance and perception by the community is assessed and reported. |
| 12 | Ramaiah, 2001 [31] | India | ComDT | Outcomes | ComDT vs traditional HS |
| 13 | Rajendran, 2010 [32] | India | Longitudinal survey with community empowerment | Outcomes | Comparisons of before & after each MDA, Prior to each MDA, HE campaigns with community as the leading player. |
| 14 | Rojanapanus, 2019 [33] | Thailand | Epidemiological survey | Structure; Process; | Baseline epidemiological survey followed by MDA. The population is divided as implementation unit (IUS) which is sub-village to ensure smaller population size, to achieve better social mobilization efforts, to achieve better compliance interim surveys in between, to find out the prevalence. |

*(Continued)*

**Table 3.** (Continued)

| No. | Study | Country, setting | Design | Type of HSR[#] | Approach |
|---|---|---|---|---|---|
| 15 | Sunish, 2016 [34] | India, Tamil Nadu | Community-based integrated VC | Structure; Process; | a group of school children from the VC villages disseminated the message on LF elimination |
| 16 | Wynd,2007 [35] | Papua New Guinea, Misima Island | Evaluation of WHO protocol for socio-cultural data collection | Process | government-private sector initiatives; FGD key informants were prominent village members, such as ward councillors, pastors, Ward Development Committee members, teachers, elders, & Women's Fellowship leaders. |

#: For more details, please refer to Table 2.

BF: bancroftian filariasis; ComDT: community-directed treatment; CP: community participation; FPAs: Filaria Prevention Assistants; GP: general practitioner; HE: health education; HS: health services; HSR: health services research; LCT: lymphoedema care team; LF: lymphatic filariasis; MDA: mass drug administration; MO: medical officers'; Outcome: Outcome of health services PHC: primary health centre; Process: Process of health services; QoL: quality of life; Structure: Structure of health services; VC: vector control.

## Who were the community and how do they interact?

Our findings showed various forms of the community involved in research on the elimination of lymphatic filariasis include (but not exhaustively) village leaders, VHW, pastors, teachers

### Country

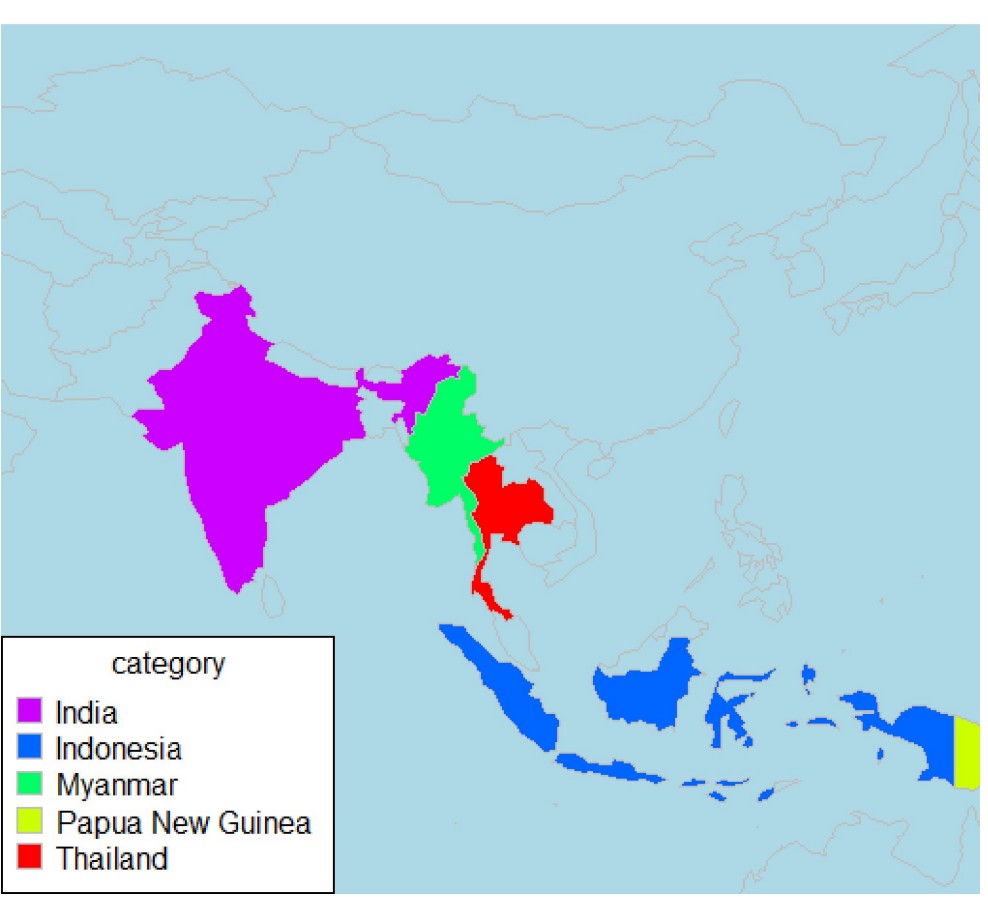

**Fig 2. Geographic distribution of studies included in the review.**

| No | Study, yr | Forms of community | | | | | | Extent of engagement in HSR | | | | | Extent of engagement | |
|---|---|---|---|---|---|---|---|---|---|---|---|---|---|---|
| | | A | B | C | D | E | F | Developing ideas | Developing methodology | Data collection/analysis | Report writing | dissemination | Scoring | level |
| 1 | Aggithaya, 2013 [21] | √ | | | | | √ | red | red | red | red | green | 0+0+0+0+1=1 | Low |
| 2 | Aye, 2018 [22] | √ | | | | | | red | green | green | red | green | 0+1+1+0+1=3 | Moderate |
| 3 | Babu, 2004 [23] | √ | | | | | | red | green | red | red | green | 0+1+0+0+1=2 | Moderate |
| 4 | Babu, 2006 [24] | √ | | | | | | red | green | green | red | green | 0+1+1+0+1=3 | Moderate |
| 5 | Dickson, 2018 [25] | | | √ | √ | | | red | green | green | red | red | 0+1+1+0+0=2 | Moderate |
| 6 | Krentel, 2016 [26] | √ | | | | | | red | green | red | red | green | 0+1+0+0+1=2 | Moderate |
| 7 | Lahariya, 2008 [27] | √ | | | | | | red | green | red | red | red | 0+1+0+0+0=1 | Low |
| 8 | Nandha, 2007 [28] | √ | | | | | | red | green | red | red | green | 0+1+0+0+1=2 | Moderate |
| 9 | Narahari, 2013 [29] | √ | | | | | | red | green | red | red | green | 0+1+0+0+1=2 | Moderate |

**Fig 3. Various forms of community and extent of engagement (N = 16 studies).**

and students, GP and political activists. Studies reported that people have individual and collective resources (time, money, materials and energy) to contribute to activities for health improvements in the community [16].

A community is commonly defined as a group of people with diverse characteristics who are linked by social ties, share common perspectives, and engage in joint action in geographical locations or settings [36]. A systematic review documented that communities are a vital and vibrant part of health systems by forming the social boundaries that define the individuals

and households whose health outcomes matter as a health systems goal, but also the social context for the relationships that underpin the success of many health systems interventions [37].

As the community and the researchers continued to meet, they developed an understanding of implementation challenges regarding the role of communities and the meaning of specific cultures [38]. However, our review could not find sufficient information on the interaction among various communities beside small meetings. Hence, it is limited to provide information of the engagement and very little, on the actual engagement of communities in the research process. From the health system perspective, a rich body of literature reported the benefits of CE/community participatory in health (CPH). These included that CE in health services can 'give a voice to the voiceless' [7], and gain information, skills and experience in community involvement that help them take control over their own lives and challenge social systems that have sustained their deprivation [16], among others. The current review is limited to find evidence on these benefits.

Similarly, a published systematic review on CE in school health research reported that none of the studies included had a singular focus on CE, rather the studies had assessed multiple health related outcomes [39]. Overall, there is a need to better understand the role of CE in HSR on elimination of lymphatic filariasis.

A systematic review on CE to reduce inequalities in health reported that interventions utilizing CE had considerable variation across populations, intervention types and outcomes [40]. Another systematic review on CE aspect of infectious diseases highlighted that the engagement of community was instrumental to increase participation in or acceptance of an intervention in the first case, and activities that foster empowerment and focus on inequalities in the second [41]. Of note, the limited reporting of outcomes observed in this review does not necessarily mean the CE is undesirable as the absence of evidence of effect does not mean the absence of an effect. A systematic review reported a lack of valid instruments for documentation in outcome assessment in the community-based participatory research [42]. Moreover, limitations to utilizing the WHO definition of drug coverage [25], and dispensing of loose tablets with no labelling on a package [22] were technology-related issues that hindered community involvement in research context.

## Study limitations

This review has some limitations that should be acknowledged. The studies included were not primarily designed to study the actual CE process in the HSR context. Hence, we used such information by extrapolating from CE in the intervention context to CE in the research process. As there were inadequate descriptions of their engagement described in the primary studies it raises concern about the accuracy of information described in the current analysis. Hence, there is a likely issue of over/under-estimation of the extent of real CE in the HSR.

In Asia, the largest subnational variations of filariasis in the (burden) estimates were predominantly in areas in Indonesia, as well as in Papua New Guinea and Myanmar [4]. Hence, the generalizability of the current findings to the LMICs context is limited by the large proportion of Indian studies in the light of cultural differences and differences in the health and social care systems.

Nevertheless, this review is the first to analyse how communities engaged in HSR on elimination of lymphatic filariasis. This may provide a guide for future implications of HSR dimension to assess CE in elimination of lymphatic filariasis. As there is no published standard validated tool to evaluate this kind of study, a methodological approach used in this study may be, to a certain extent, applicable to other disease interventions (e.g dengue infection, soil-transmitted helminths) in the LMIC context with necessary modifications

### Implications

Based on the scarcity of research on the extent of CE in HSR process in this systematic review, there is a need for studies to document: (i) a definition of "community" in CE in research, (ii) develop conceptual models for measuring outputs and outcomes from engaging community in the research [40], and (iii) apply the lessons learned from such studies more broadly across HSR. It has been highlighted that such engagement needs to occur as the ideas for research are being formed and the procedures are being identified, by taking the community's priorities into account and being a regular presence in the community may enhance research efforts [9].

### Conclusions

The findings suggest there was limited community engagement in HSR on lymphatic filariasis elimination. Future well-designed prospective studies addressing communities' engagement in research focusing on the elimination context of lymphatic filariasis and more attention to the evaluation schemes emphasizing the description of mechanisms of engagement, facilitating the whole process and reporting the community level outcome using an adjustment by combining qualitative quantitative methods are recommended.

### Supporting information

**S1 Checklist. PRISMA checklist.**
(DOC)

**S1 Text. R commands for mapping Fig 2.**
(DOC)

**S1 Table. Search strategy.**
(DOC)

**S2 Table. Excluded studies.**
(DOC)

**S3 Table. Summary of risk of bias assessment using ROBINS-1 tool.**
(DOC)

**S4 Table. Barriers and facilitators to community engagement in HSR.**
(DOC)

### Acknowledgments

We thank participants and researchers involved in the primary studies identified for this review, and our institutions for allowing us to perform this study. We are very grateful to the editor and reviewers for providing comments and valuable inputs to improve the quality of our manuscript.

### Author Contributions

**Conceptualization:** Cho Naing, Maxine A. Whittaker.

**Data curation:** Cho Naing, Norah Htet Htet, Htar Htar Aung.

**Formal analysis:** Cho Naing, Norah Htet Htet, Maxine A. Whittaker.

**Funding acquisition:** Cho Naing, Norah Htet Htet, Htar Htar Aung, Maxine A. Whittaker.

**Investigation:** Norah Htet Htet, Maxine A. Whittaker.

**Methodology:** Cho Naing, Norah Htet Htet, Htar Htar Aung, Maxine A. Whittaker.

**Project administration:** Cho Naing, Maxine A. Whittaker.

**Writing – original draft:** Cho Naing, Maxine A. Whittaker.

**Writing – review & editing:** Norah Htet Htet, Htar Htar Aung.

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
