## [Decision Letter · Decision Letter 0]

4 Oct 2022

PGPH-D-22-01335

Community Engagement in Health Services Research on Elimination of Lymphatic Filariasis: A Systematic Review

Dear Dr. Naing,

Thank you for submitting your manuscript to PLOS Global Public Health. After careful consideration, we feel that it has merit but does not fully meet PLOS Global Public Health’s publication criteria as it currently stands. Therefore, we invite you to submit a revised version of the manuscript that addresses the points raised during the review process.

We look forward to receiving your revised manuscript.

Kind regards,

Giridhara R Babu, MBBS, MPH, PhD

Academic Editor

Journal Requirements:

3. We do not publish any copyright or trademark symbols that usually accompany proprietary names, eg (R), (C), or TM  (e.g. next to drug or reagent names). Please remove all instances of trademark/copyright symbols throughout the text, including ® on pages 9.

Additional Editor Comments (if provided):

Reviewers' comments:

Reviewer's Responses to Questions

**Comments to the Author**

1. Does this manuscript meet PLOS Global Public Health’s publication criteria? Is the manuscript technically sound, and do the data support the conclusions? The manuscript must describe methodologically and ethically rigorous research with conclusions that are appropriately drawn based on the data presented.

Reviewer #1: Yes

Reviewer #2: Yes

2. Has the statistical analysis been performed appropriately and rigorously?

Reviewer #1: N/A

Reviewer #2: Yes

3. Have the authors made all data underlying the findings in their manuscript fully available (please refer to the Data Availability Statement at the start of the manuscript PDF file)?

Reviewer #1: Yes

Reviewer #2: Yes

4. Is the manuscript presented in an intelligible fashion and written in standard English?

Reviewer #1: Yes

Reviewer #2: Yes

5. Review Comments to the Author

Reviewer #1: The authors did a good work on the systematic review. Please find my concerns as comments:

Abstract:

1. The last sentence reads incomplete. Are "Future well-designed prospective studies..." recommended?

Introduction:

1. Line 68, write out CE at first instance, or insert acronym on Line 65 if it means "Community Engagement.

Methods:

1. Authors listed some databases searched for articles, but I could not find details in the supplementary S1 table such as for Web of Science, Science Direct, Google scholar, WHO library. Kindly clarify.

2. Was there a justification or cited standardization for categorizing overall rating as high (score 4-5), moderate (score 3) and low (score ≤ 2)? Why, for example, was moderate not (score 3-2) and low (score 1-0)?

3. If the investigators reviewed 'full-text copies' (line 142), should not your exclusion criteria include inaccessibility of full-text articles? Else, state that you were able to retrieve full-text copies of all the articles from your search strategy.

Results:

Looks good.

I would recommend line 230 - 234 be moved to discussion section. "For instance, limitations to utilizing the WHO definition of drug coverage..."

Figure 1 PRISMA flow: Records identified (1069) minus Duplicated records (143) leave Record screened (926). This is correct. Records removed for other reasons (378) disrupts this flow, because your diagram implies that 378 was removed from 1069, which was not the case. Please modify diagram to show where 378 was removed.

Minors:

Write out ROBINS-I at first instance: "Risk Of Bias In Non-randomised Studies - of Interventions"

Did you consider "Stakeholders involvement" as keyword in your search strategy?

Reviewer #2: Summary

Thank you for the opportunity I have to provide a review for this Community Engagement in Health Services Research on Elimination of Lymphatic Filariasis: A Systematic Review.

This is a good attempt by the authors to determine engagement of the community in health research and the conclusion showed there is need to adequate information of process of community engagement in these researches that were reviewed.

Title

• The title clearly depicts the research that was carried out with some focus on what is being reviewed, however the period or time of investigation is missing. I think it should be included.

• Each author’s contribution was clearly stated, and corresponding author’s contact is clear.

Abstract

• The abstract is unstructured, and it allowed for a good read.

• The conclusion was clearly written

Introduction

• The introduction provided a good background for the study.

• line 68: authors introduced an acronym (“CE”) that was not explained before. Please write it in full first so readers can understand what it is for.

• The objectives of the study were stated clearly.

Methods and Materials

• methods and materials are well explained, and it is reproducible

Results

• The result is well written

Discussion

• The discussion is focused on the results

Conclusion

• Good conclusion

Recommendation

• Accept

6. PLOS authors have the option to publish the peer review history of their article (what does this mean?). If published, this will include your full peer review and any attached files.

**Do you want your identity to be public for this peer review?** For information about this choice, including consent withdrawal, please see our Privacy Policy.

Reviewer #1: **Yes: **Ruxton Adebiyi

Reviewer #2: **Yes: **ADEBAYO PETER ADEWUYI

---

## [Decision Letter · Decision Letter 1]

1 Nov 2022

PGPH-D-22-01335R1

Community Engagement in Health Services Research on Elimination of Lymphatic Filariasis: A Systematic Review

Dear Dr. Naing,

Thank you for submitting your manuscript to PLOS Global Public Health. After careful consideration, we feel that it has merit but does not fully meet PLOS Global Public Health’s publication criteria as it currently stands. Therefore, we invite you to submit a revised version of the manuscript that addresses the points raised during the review process.

**Please address the review Comments to the Author and submit revised manuscript.**

We look forward to receiving your revised manuscript.

Kind regards,

Giridhara R Babu, MBBS, MPH, PhD

Academic Editor

Journal Requirements:

Additional Editor Comments (if provided):

Reviewers' comments:

Reviewer's Responses to Questions

**Comments to the Author**

1. If the authors have adequately addressed your comments raised in a previous round of review and you feel that this manuscript is now acceptable for publication, you may indicate that here to bypass the “Comments to the Author” section, enter your conflict of interest statement in the “Confidential to Editor” section, and submit your "Accept" recommendation.

Reviewer #2: All comments have been addressed

Reviewer #3: All comments have been addressed

Reviewer #4: All comments have been addressed

2. Does this manuscript meet PLOS Global Public Health’s publication criteria? Is the manuscript technically sound, and do the data support the conclusions? The manuscript must describe methodologically and ethically rigorous research with conclusions that are appropriately drawn based on the data presented.

Reviewer #2: Yes

Reviewer #3: Partly

Reviewer #4: Yes

3. Has the statistical analysis been performed appropriately and rigorously?

Reviewer #2: Yes

Reviewer #3: Yes

Reviewer #4: Yes

4. Have the authors made all data underlying the findings in their manuscript fully available (please refer to the Data Availability Statement at the start of the manuscript PDF file)?

Reviewer #2: Yes

Reviewer #3: Yes

Reviewer #4: Yes

5. Is the manuscript presented in an intelligible fashion and written in standard English?

Reviewer #2: Yes

Reviewer #3: Yes

Reviewer #4: Yes

**6. Review Comments to the Author**

Reviewer #2: No new comments, all previous comments duly addressed

Reviewer #3: The title should follow the PLOS requirements; it should be written in sentences case with its exceptions. Please revise it.

The corresponding author should not be written in abbreviation.

The sign for "These authors contributed equally" is not the correct sign. Please revise it.

Line 80: After a semi colon the first word should be written in capital letter

Line 240: Why full stop? I think you want to make a coma. please update it.

Line 244-245: The statement needs a coma.

After the headings, you have started the text with out paragraph indentation. However, after the study limitation headings (line 290) you have started with the indentation.

Reviewer #4: The manuscript has been impressively improved based on previous reviewer comments. There are few areas for improvement which have been given below:

1. In outcome section of table 1, give operational definition of all the outcomes listed.

2. Lines 189-190- please briefly explain HSR typology (structure, process and outcome) in methodology section.

3. In table 3, study number 5, there are two green bubbles are shown, but the score is just 1, this should be corrected.

4. In table 3, description of F form of community is missing.

7. PLOS authors have the option to publish the peer review history of their article (what does this mean?). If published, this will include your full peer review and any attached files.

**Do you want your identity to be public for this peer review?** For information about this choice, including consent withdrawal, please see our Privacy Policy.

Reviewer #2: **Yes: **ADEBAYO PETER ADEWUYI

Reviewer #3: No

Reviewer #4: **Yes: **Sudeep Adhikari

---

## [Decision Letter · Decision Letter 2]

13 Dec 2022

Community engagement in health services research on elimination of lymphatic filariasis: A systematic review

PGPH-D-22-01335R2

Dear Cho Naing

We are pleased to inform you that your manuscript 'Community engagement in health services research on elimination of lymphatic filariasis: A systematic review' has been provisionally accepted for publication in PLOS Global Public Health.

Best regards,

Janelisa Musaya, PhD

Academic Editor

Reviewer Comments (if any, and for reference):

Reviewer's Responses to Questions

**Comments to the Author**

1. If the authors have adequately addressed your comments raised in a previous round of review and you feel that this manuscript is now acceptable for publication, you may indicate that here to bypass the “Comments to the Author” section, enter your conflict of interest statement in the “Confidential to Editor” section, and submit your "Accept" recommendation.

Reviewer #3: All comments have been addressed

Reviewer #4: All comments have been addressed

2. Does this manuscript meet PLOS Global Public Health’s publication criteria? Is the manuscript technically sound, and do the data support the conclusions? The manuscript must describe methodologically and ethically rigorous research with conclusions that are appropriately drawn based on the data presented.

Reviewer #3: Yes

Reviewer #4: Yes

3. Has the statistical analysis been performed appropriately and rigorously?

Reviewer #3: Yes

Reviewer #4: I don't know

4. Have the authors made all data underlying the findings in their manuscript fully available (please refer to the Data Availability Statement at the start of the manuscript PDF file)?

Reviewer #3: Yes

Reviewer #4: Yes

5. Is the manuscript presented in an intelligible fashion and written in standard English?

Reviewer #3: Yes

Reviewer #4: Yes

6. Review Comments to the Author

Reviewer #3: Please include "These authors contributed equally" with the sign indicated.

Reviewer #4: All comments addressed properly.

7. PLOS authors have the option to publish the peer review history of their article (what does this mean?). If published, this will include your full peer review and any attached files.

**Do you want your identity to be public for this peer review?** For information about this choice, including consent withdrawal, please see our Privacy Policy.

Reviewer #3: **Yes: **Eyob Girma Abera

Reviewer #4: **Yes: **Sudeep Adhikari
